# Biomechanical Properties of Cancer Cells

**DOI:** 10.3390/cells10040887

**Published:** 2021-04-13

**Authors:** Gaël Runel, Noémie Lopez-Ramirez, Julien Chlasta, Ingrid Masse

**Affiliations:** 1Centre de Recherche en Cancérologie de Lyon, CNRS-UMR5286, INSREM U1052, Université de Lyon, F-69008 Lyon, France; gael.runel@bio-meca.com (G.R.); noemie.lopez-ramirez@univ-lyon1.fr (N.L.-R.); 2BioMeca, F-69008 Lyon, France; julien.chlasta@bio-meca.com

**Keywords:** cancer, biomarker, biomechanics

## Abstract

Since the crucial role of the microenvironment has been highlighted, many studies have been focused on the role of biomechanics in cancer cell growth and the invasion of the surrounding environment. Despite the search in recent years for molecular biomarkers to try to classify and stratify cancers, much effort needs to be made to take account of morphological and nanomechanical parameters that could provide supplementary information concerning tissue complexity adaptation during cancer development. The biomechanical properties of cancer cells and their surrounding extracellular matrix have actually been proposed as promising biomarkers for cancer diagnosis and prognosis. The present review first describes the main methods used to study the mechanical properties of cancer cells. Then, we address the nanomechanical description of cultured cancer cells and the crucial role of the cytoskeleton for biomechanics linked with cell morphology. Finally, we depict how studying interaction of tumor cells with their surrounding microenvironment is crucial to integrating biomechanical properties in our understanding of tumor growth and local invasion.

## 1. Introduction

In recent decades, much effort has gone into deciphering molecular mechanisms and identifying biomolecular markers to help to stratify cancers and predict local invasion and metastasis. Indeed, the emergence of genomic profiling technologies has allowed for the identification of some prognostic and predictive biomarkers and has improved the clinical management of some cancer patients. More than 69,000 publications are indexed in PubMed with subject headings “prognostic marker” and “neoplasm”, and some are used by clinicians to make treatment decisions. For instance, in breast cancer, multiple gene/protein signatures have been proposed to predict outcome and even response to therapy (reviewed in [1]). More recently, extended research identified other types of biomolecular markers such as microRNAs and long ncRNAs that could serve as new molecular tools for diagnosis and prognosis in various cancers (reviewed in [2]). Ongoing analysis of circulating tumor cells (CTCs), collected from the patient’s blood, demonstrated that these could help to predict metastatic potential and to monitor treatment response (reviewed in [3]). However, molecular studies only are not sufficient to understand cancer progression and response to therapies and there is growing evidence of the role of tumor microenvironment in the development and evolution of various cancer types. The microenvironmental contribution was shown in the acquisition of genomic instability of malignant solid tumors and multiple myelomas (reviewed in [4]). Recent decades were also marked by a growing interest in onco-immunology to decipher the role of the immune system in cancer initiation and progression and resistance to therapy (reviewed in [5]). However, although tumor stratification was improved by these recent discoveries, pathologic analysis remains of prime importance in patient care. The recent development of more precise techniques for studying morphological changes in tumor cells and biomechanical characteristics inside the tumor and in the surrounding microenvironment is promising for tumor detection and classification. The present review describes some of the main findings regarding the ability of cancer cells to adapt their morphology and invasiveness by remodeling their cytoskeleton in connection with their nanomechanical properties. After a brief introduction to the mechanoreciprocity between tumor cells and their microenvironment in terms of biochemical and biomechanical cues, we list the main methods of biomechanical analysis of cancer cells. Then, we describe studies showing how various nanomechanical cell parameters can discriminate cancer cells from healthy ones, and the role the cytoskeleton plays in cancer cell morphology and invasiveness. Finally, we highlight the importance of taking account of the crucial role of the microenvironment in determining the biomechanical characteristics of tumor cells.

## 2. Interplay between Biochemical and Biomechanical Features of Tumor Cells in Their Microenvironment

A tumor in expansion inside the surrounding normal tissue will inevitably destabilize the tissue homeostasis. The development of cancer leads to changes in cellular biomechanics both for cancer cells and for cells of the tumor microenvironment. These changes translate concomitantly into specific biomechanical cues. It is well known that tumors become stiff due to fibrosis during tumor progression, and this impacts the biochemical and physical properties of tumor cells. Important hallmarks of cancer can be impacted by these changes, such as increased cell proliferation, resistance to cell death, increased cell motility and invasiveness, or induction of angiogenesis (for reviewed in [6]). Stiffening of the extracellular matrix (ECM) is also observed. Cancer-associated fibroblasts (CAFs) tumor cell migration by modifying collagen reticulation [7,8,9]. These changes in the mechanical properties of the tumor microenvironment due to tumor growth lead to the activation of molecular pathways: i.e., the β-catenin and Yap/Taz signaling pathways, the role of which in mechanotransduction has been well described in both tumor cells and CAFs [6,10]. Mechanosensing of ECM stiffness by cancer cells and its translation into biochemical cues have been attributed to various players, such as junctional and membrane proteins, cytoskeleton and nucleoskeleton and more recently to microRNAs [6].

However, the techniques for analyzing biomechanical properties are relatively recent and it is still difficult to study the biomechanical properties of cancer cells in the context of the microenvironment in order to take account of the mechanoreciprocity between tumor cells and the surrounding ECM. Thus, most experiments have so far been conducted on cultured cancer cells and mostly focused on cytoskeleton properties, since drugs targeting the cytoskeleton are easy to apply. We will thus mainly describe the techniques relating to the mechanical properties of cancer cells and their reciprocal correlation with cytoskeletal and morphological properties. We will, however, finally discuss the few recent studies that sought to include mechanoreciprocity between tumor cells and the microenvironment in vivo.

## 3. Main Methods to Study Mechanical Properties of Cancer Cells

Since the late 1990s, various new technologies have enabled the measurement of different biophysical parameters such as cell stiffness, viscoelasticity or deformability, shedding light on the mechanics of cancer cells. These are presented in Figure 1 with their respective properties and principles.

### 3.1. Atomic Force Microscopy

Atomic force microscopy (AFM) is especially useful for single-cell analysis of the properties of cultured adherent cells. Three physicists, Gerd Binnig, Calvin Ouate and Christoph Gerber, invented the AFM technology in 1985 [11]. It is a microscopy technique using a type of scanning probe microscope, based on a cantilever able to contact the sample and scan it directly. AFM is particularly suitable for studying adherent cells and tissues. The resolution limit of AFM imaging depends on the geometry of the probe tip, which measures a few nanometers and is fixed at the free end of a cantilever, able to move in XYZ directions to scan the sample. To record the Z tip position, a laser beam is focused on the free end of the cantilever, just above the probe tip. The reflected beam is directed toward a four-quarter photodiode. Depending on the cantilever’s mechanical properties, the applied force and the position of the laser beam on the photodiode, the mechanical and topographical characteristics of the sample are extracted. The very high resolution of AFM is achieved through the small size of the tip, measuring just a few nanometers, capturing very small samples such as DNA [12] or period collagen [13]. A large range of cantilevers is available, with a spring constant from 0.06 N/m to 200 N/m for cell and tissue measurement, with conical or spherical tips.

AFM is able to extract many physical parameters from a sample. In contact mode, the probe tip is in constant physical contact with the sample surface, and the Z piezo motor moves along the surface, keeping a constant force applied by the cantilever. AFM gives the XYZ position for each pixel, reconstructing the topography of the sample [14]. In tapping mode, the cantilever oscillates up and down (or near its resonance frequency) near the sample surface. The cantilever tip is thus partially in contact with the sample. Each pixel corresponds to a force indentation curve, determining the topography, by extracting the contact point on the indentation curve, and mechanical characteristics of the sample (Figure 2a) [15]. Mechanical properties such as stiffness or viscoelasticity [16] as well as electric properties [17] can be measured by AFM, and theoretical models (Hertz or Sneddon, for instance) are used to calculate quantitative parameters, and in particular the Young’s modulus, which represents the measure of stiffness and resistance to elastic deformation [18]. Viscoelasticity is measured by successive hold indentations and oscillations at normalized frequency, followed by a final retraction of the indenter [19]. The hysteresis between the approach and the retract curve determines the viscosity or plastic behavior of the sample. Plastic behavior is stiffer and deformable under high constraint, easily returning to its original shape, whereas a viscous sample is softer and more deformable (Figure 2b).

### 3.2. Micropipette Aspiration, Optical Stretcher and Microfluidic Systems

Other techniques are available to extract cell deformability and viscosity. The oldest is micropipette aspiration (MPA), where a known negative pressure is applied on the cell by micropipette and a vacuum device [20,21]. Negative pressure is applied at the tip of a glass micropipette and, when the cell is aspirated into the micropipette, progressive deformation can be measured (Figure 2c). With a known negative pressure and a high-resolution camera, deformability can be extracted, and mathematical models quantify the viscoelastic properties of the sample. This MPA technique is particularly suitable for non-adherent cells, in contrast to AFM.

Techniques based on the capacity of a laser to trap and stretch particles by photon momentum effect, deforming cells in suspension, have also been developed. The optical stretcher (OS) uses two focalized laser beams that apply about 100 pN force to deform the sample. One particularity of OS is that it is completely contact-free. This technique can be combined with microfluidic cytometry [22].

The recent development of microfluidic approaches has improved the measurement of cell viscoelasticity and deformability. Microfluidic systems are based on a small chip where cells cross tight capillaries under flow effect at high speed. A high-resolution camera extracts the deformability properties of the cells and the velocity at which they can circulate through the microfluidic channels [23]. These techniques screen a large number of cells with very high throughput.

Taken together, these various methods for studying biomechanical phenotypes of individual cells or tissues can characterize them in terms of stiffness, viscoelasticity and deformability (Figure 1 and Figure 2). They have been used extensively in recent years, especially to analyze the nanomechanical features of cancer cells.

## 4. Stiffness, Viscoelasticity and Deformability of Cultured Cancer Cells

### 4.1. Pioneering Experiments on the Biomechanics in Cancer Cells

Using the techniques described above, numerous studies were performed on 2D-cultured cells, at various stages of cancer development. They showed the systematic modulation of stiffness during tumoral transformation and progression. AFM in a mouse embryonic model [24] for the first time enabled the study of the viscoelastic properties of individual mouse F9 embryonic carcinoma cells [25]. Shortly afterward, comparison by MPA between normal and transformed fibroblasts showed differences in mechanical properties, especially in terms of intrinsic resistance of the cellular structure and deformation capacity [26]. In 1999, Lekka et al. conducted a more detailed study using AFM to study the elastic properties of normal ureter or bladder cell lines (Hu609 and HCV29 respectively) versus cancerous bladder cell lines (Hu456, T24, BC3726) [27]. Interestingly, BC3726 cells were directly derived from HCV29 non-malignant bladder urothelial cells after transformation by the V-ras oncogene, allowing comparison between healthy and paired-transformed phenotypes. The authors showed that the mechanical force required for indentation was greater for normal than for cancerous cells, with a significant 30-fold decrease in Young’s modulus between normal and cancerous cell lines. These data confirmed the original experiments conducted on resuspended normal versus transformed human dermal fibroblasts [26,28].

### 4.2. Stiffness and Viscoelasticity of Cancer Cells

In the 2000s, numerous studies in various types of cancer strengthened the evidence for the modulation of cell nanomechanical features, such as stiffness, viscoelasticity and deformability, with cell transformation and cancer progression. In breast cancer especially, analysis of the benign MCF10A, premalignant MCF10AT and invasive malignant MCF10CA1A, MDA-MB-231 and MCF7 cell lines showed reduction in stiffness and viscoelasticity: i.e., breast cancer cells were softer and more fluid than their benign counterparts [29,30,31,32,33]. AFM measurements were also conducted in melanoma cell lines representative of different stages of tumor progression, from normal human epidermal melanocytes (NHEM) to metastatic melanoma cells (WM239A), with intermediate non-invasive and locally invasive cell lines (SBC12 and WM115). Initial stages of transformation correlated with decreasing stiffness from normal melanocytes to locally invasive melanoma cells, whereas the fully metastatic cell line presented higher and more heterogeneous stiffness values. The authors hypothesized that melanoma cells, well known for their plastic abilities, could adapt their stiffness along tumor progression so as to response optimally to the microenvironment [34]. Similar experiments compared other bladder [35], melanoma [36], prostate [37,38,39], chondrosarcoma [40], ovarian [41,42], urothelial [43], liver [44], esophageal [45], cervical [46] and thyroid [47] cancerous cell lines, versus their non-malignant counterparts. Interestingly, reverse experiments were performed in SH-SY5Y human neuroblastoma cells. These undifferentiated tumoral cells can be differentiated into neuron-like cells by retinoic acid and Brain-Derived Neurotrophic Factor (BDNF) treatment. The differentiation caused by the chemical treatment induced functional changes in SH-SY5Y cells, leading to the loss of some characteristic features of cancer cells. These functional changes were accompanied by changes in biomechanical properties: differentiated cells became three times stiffer than undifferentiated tumoral SH-SY5Y cells [48,49].

### 4.3. Deformability Properties of Cancer Cells

The modification of cancer cell deformability, inversely correlated with stiffness, was more specifically demonstrated by OS techniques in cell cultures in suspension. Guck et al. were the first to show a link between deformability and the aggressiveness of breast cancer cells, using cell lines with increasingly invasive properties [50]. They first demonstrated, in mouse fibroblasts, that normal cells were less deformable than their counterparts transformed with the oncogenic DNA virus SV40. Moreover, the two populations were easily distinguishable by measuring this biomechanical characteristic. They also showed that optical deformability of the different cell lines could be solely attributed to their mechanical properties. By performing the same type of experiments on human breast samples, they demonstrated that metastatic cell lines were more deformable than non-metastatic or non-tumoral cell lines [50,51]. The increased deformability of tumor cells was then confirmed by comparing normal versus cancerous oral epithelial cell lines [52] and normal versus Ras-transformed epithelial cells [53]. In the late 2010s, the development of large-scale micro-fluidic technologies confirmed and extended these conclusions in a large panel of cancer cell types. Cancer stem-cell mechanomics and cellular deformation were studied in glioblastoma [54], breast cancer [55] and osteosarcoma [56] by real-time deformation cytometry (DC); increasing cell deformability with the progression of breast and prostate cancer was demonstrated in cell lines with growing invasiveness by inertial microfluidic cell stretcher (iMCS) and multi-sample DC [23,57]; and metastatic and non-metastatic breast cell lines were able to be distinguished by mechanical separation chips (MS-chip) based on differential deformability [51]. Overall, in a broad range of cancer cell types, biomechanical measurements could be considered as biomarkers of both cell transformation and tumor progression.

### 4.4. Ex-Vivo Cancer Cell Analyses

Interestingly, biomechanical properties have also been studied ex vivo, in patient samples, with promising results in terms of nanomechanical biomarker characterization. Metastatic cancer cells from pleural effusions of patients suffering from lung, breast and pancreas cancer were 70% to 80% softer that their benign counterparts [58,59,60]. Using DC, Tse et al. showed that disseminated tumor cells from malignant pleural effusion could be distinguished on mechanophenotype as malignant or negative, in agreement with the observed 6-month outcome of the patients [61,62]. Finally, circulating CTCs can also be discriminated on biomechanical parameters: CTCs from castration-resistant prostate cancer patients were three times softer and more deformable than their castration-sensitive counterparts [63].

## 5. Cancer Cell Cytoskeleton, Cell Morphology and Biomechanical Properties

An important component of cell mechanics is unquestionably the cytoskeleton. It determines cell morphology, maintains mechanical integrity, contributes to resistance to external forces, and can impact both mechanotransduction and migration properties. Variation in the concentration of the various molecules composing the cytoskeleton and their compartmentalization and post-translational modifications influence its mechanical properties and its capacity to deform in order to adapt its morphology in response to microenvironmental chemical and/or physical stimuli. The three elements of the cytoskeleton are microtubules, microfilaments and intermediate filaments. They are mainly composed of tubulin, F-actin and vimentin/desmin/keratin proteins, respectively. Their individual concentrations in the cell and their restructuration ability are essential for the different cell activities. Although few studies had been conducted at that time, in 1991 P.A. Janmey reviewed the physical properties of these biopolymers and how in vivo remodeling affects cell shape and motility [64]. By playing with various concentrations of these biopolymers (F-actin, for example) and applying deformation until the filaments broke, various studies showed that fiber diameter and inter-fiber interactions are crucial for cell dynamics and physical properties. Some differences appeared on measurements of purified networks, especially by AFM imaging. For instance, the actin network is important for cell biomechanics and adaptability to very low mechanical stress. On the contrary, the disruption of intermediate filaments revealed smaller effects on cell stiffness, suggesting that intermediate filaments are involved more in resistance to stronger mechanical forces, impacting cell migration, adhesion and mechanotransduction (reviewed in [65]). Consequently, nanomechanical measurements of cells in culture mainly reveal the reorganization of the actin network. Some biomechanical studies conducted in normal adherent or non-adherent cells (fibroblasts or macrophages) clearly demonstrated that modifying the actin network and/or microtubule polymerization by drugs such as cytochalasins [66] or docetaxel led to nanomechanical changes in the cell [67,68,69,70,71,72].

### 5.1. Cytoskeleton Disruption by Drugs and Biomechanics in Cancer Cells

Despite the discoveries regarding the cytoskeleton’s important roles in cell shape and biomechanical properties, few studies have used drugs able to modulate the cytoskeleton in cancer cell cultures. For example, the treatment of cervical cancer HeLa cells by cytochalasin B, which blocks actin polymerization and elongation without affecting the structure of microfilaments or intermediate filaments, significantly decreased the cellular elastic modulus: i.e., cells became softer when F-actin was damaged [73]. Cytochalasin D treatment also reduced the Young’s modulus of human bladder cancer cells and increased cellular deformability [35,74]. This suggested that the viscoelastic properties of these bladder carcinoma cell lines could be mainly attributed to the 3D-organization of actin filaments. These observations could be juxtaposed with the basal differences in the actin network of grade II and III bladder carcinoma cells compared to non-malignant or transitional bladder cells. In the two first malignant cell lines, only short actin filaments were observed whereas normal bladder cells showed actin fibers organized in both short filaments and bundles of long filaments named stress fibers [35]. These findings highlighted changes in the actin cytoskeleton structure during cancer progression, reflected by cell stiffness, which could thus be a powerful biomarker, especially in human bladder tumors. Similar results were later observed in breast cancer cells [75,76].

The same kind of experiments were performed with drugs affecting the polymerization and stability of the microtubule network. Paclitaxel treatment, which stabilizes microtubules, decreased cell stiffness in Ishikawa cells and Hela cells [77] but increased Young’s modulus in melanoma cells [78]. Treatment by colchicine or docetaxel drugs, which induce functional disturbances in the microtubule network, modified the elastic modulus in prostate cancer cells [79] and in some hepatoma cells, without affecting normal hepatocytes [80].

Overall, these data showed that playing directly on the cytoskeleton with drugs that directly target microtubules or actin microfilaments deeply affects cell structure and biomechanical parameters. The less constant effect seen with microtubule-disturbing drugs could reflect the fact that, as in normal cells, nanomechanical measurements of tumoral cells in culture are more profoundly affected by reorganization of the actin network than by the microtubule network [81]. Very few studies have been carried out on the role of intermediate filaments, but it appears that keratin reorganization in human pancreatic cancer cells and transformed keratinocytes alters their stiffness and viscoelastic properties [82,83]. Similarly, spatial reorganization of the vimentin intermediate filament network by the overexpression of various oncogenes in human skin fibroblasts modulated their stiffness [84].

### 5.2. Link between Cytoskeleton, Morphology and Nanomechanical Properties of Cancer Cells

Other studies linked cytoskeleton disruption not only to nanomechanical modulation (stiffness, elasticity, deformability) but also to cell morphology and invasiveness. In mouse fibroblasts and human breast epithelial cells, malignant transformation was associated with the reduction in and reorganization of cytoskeleton filamentous actin, coupled with an increase in cell deformability [50]. Fibroblast transformation was also associated with lamellipodia remodeling, actin reorganization and reduced stiffness [85]. Interestingly, when highly metastatic MDA-MB-231 breast cancer cells were treated by *trans* retinoic acid to reduce their aggressive phenotype, a decrease in their deformability was observed in correlation with the modulation of cytoskeletal resistance to deformation. Conversely, the disruption of the actin or microtubule networks in these breast cancer cells changed their shape and increased cell deformability [23]. Also, the inhibition of microtubule synthesis and actin filament disruption by fullerenol [86,87] significantly decreased the stiffness of human hepatocellular carcinoma cells and diminished long actin filaments, transforming them into actin aggregates irregularly distributed in the cells, which became retracted and acquired a rounder morphology [88]. Likewise, when SH-SY5Y human neuroblastoma cells were treated with N-methyl-D-aspartate (NMDA), known to regulate intracellular Rho GTPases, which contribute to forming bundles of long actin filaments [89], stiffness measured by AFM was disturbed, as cell surface and neurite extensions were modified [90]. The importance of Ras/Rock (Rho Associated Coiled-Coil Containing Protein Kinase) GTPase-dependent cytoskeleton organization in stiffness modulation was also shown in epithelial transformed cells [53]. Finally, in melanoma cells, the modification of the actin filament distribution in cells undergoing oxidative stress was also accompanied by drastic modification in cell morphological and cell biomechanical parameters (stiffness and elasticity) [91].

Taken together, these data highlight the importance of the cytoskeleton in modulating both the morphological and the biomechanical properties of cancer cells.

## 6. Cell-Environment Mechanical Interaction in Cancer

As previously mentioned, most studies focused on the nanomechanical properties of isolated cancer cells. Only recently have the stiffness, deformability and morphology of cancer cells been studied in three dimensions, integrating the important role of tissue structure and the surrounding extracellular matrix (ECM). Even in normal cells, it is well established that growing them on ECMs of different rigidities modifies their morphology, adhesion and cytoskeletal organization as well as their stiffness [92,93], and that ECM collagen I and fibronectin can modulate mechanical forces depending upon matrix tension [94].

### 6.1. Mechanoreciprocity between Tumor Cells and the Surrounding ECM

Tumors were first analyzed as a whole, and were in general described as stiffer than the surrounding normal tissue [95,96,97,98,99], due partly to collagen crosslinking in the ECM by malignant and stromal cells [100,101]. The interplay between cellular and ECM stiffness is all the more crucial due to the mechanoreciprocity between the ECM and the expanding tumor within the surrounding normal tissue. Cell biomechanical properties may be different at the periphery and in the core of the tumor. The structure and organization of a tumor is continuously changing as the cancer evolves, and the biomechanical state of the cells that compose the tumor may evolve at the same time. Tumor cells thus need to adapt themselves to the physical pressure exerted by the microenvironment as the tumor grows and is remodeled at the periphery and in the core. The application of biomechanical stress on human breast tumors grafted into Nude mice led to modification of tumor volume over time [102]. At the cellular level, it has been shown that mimicking compression of osteosarcoma cells by confining them inside ECM-functionalized channels led to dramatic changes in cell shape and a progressive reduction in cell stiffness [103]. How the stiffness of the microenvironnement drives malignant transformation and progression was first described in breast cancer [100,104,105]. Consistently with the importance of the link between modulation of biomechanical properties and reorganization of the cytoskeleton in tumor cell cultured in vitro, even a slight change in ECM rigidity can disturb tissue architecture and favor tumor growth by regulating Rho GTPase signaling [104]. Changes in the mechanical properties of the microenvironment by cultivating cells in 3D collagen matrices were also shown to lead to changes in cell viscoelasticity in various other cancer types such as colorectal, pancreatic or prostate carcinoma [55,106].

### 6.2. Mechanoreciprocity during Local Invasion

The important role of mechanoreciprocity between tumor cells and their surrounding microenvironment was specifically investigated during invasion to try to explain why some cells with certain biomechanical properties more promptly acquire higher invasiveness. Differences in cell morphology and stiffness were shown to be more pronounced in various head and neck carcinoma cell lines differing in invasiveness when they were cultivated in 3D-collagen matrices [107]. This underlies the importance of studying cell biomechanics in an in vivo-like microenvironment to better understand invasion features. A correlation between invasive potential and a cancer cell’s ability to adjust mechanically to the surrounding ECM was clearly established in pancreatic cancer cells [55]. Recent techniques combining AFM, confocal fluorescent microscopy and finite element mathematical modeling enabled the quantitative study of the viscoelastic properties of both tumor cells and the surrounding matrix. Breast cancer cells stiffened as they invaded collagen I matrices, and this stiffening depended on Rho/ROCK signaling [108]. Cell stiffening may be mediated by transient accumulation of stress fibers prior to acquisition of malignant features that help tumor growth in premalignant stages [109]. The importance for cancer cells to become transiently stiffer may be directly linked to ECM remodeling in tumoral contexts. Profound ECM reorganization during tumor development, with densification of the collagen network and a radial alignment of collagen fibers, may be partly due to CAFs remodeling [7,8,9]. These tracks help tumor cells to migrate collectively in local invasion [110,111]. Interestingly, Kaur et al. showed that age-related ECM modification changes, with increased matrix alignment and ECM stiffness, promoted melanoma cell invasion [112]. Overall, these new observations prove the importance of the 3D tumoral microenvironment for better understanding the biomechanics underlying the behavior of cancer cells in their physiological context.

Very recently, Han et al., using organoids of mammary cancer cells in hydrogel, demonstrated that tumors exhibit heterogeneous biomechanical cell patterns that facilitate tumor invasion. Cells at the organoid periphery were softer and larger than those in the core. Stiffness became increasingly heterogeneous as the organoid developed. Moreover, eliminating the softer peripheral cells of cancer organoids delayed the transition toward an invasive phenotype [113], thus linking biomechanical properties to tumor progression toward invasiveness. These data are all the more interesting in that Gensbittel et al. recently proposed a model whereby cancer cells continuously adapt their biomechanical properties throughout the various steps of transformation and progression [114]. It can be hypothesized that cancer cell plasticity, perhaps through the modulation of the epithelial-to-mesenchymal (EMT) transition, is a way for tumor cells to be more compliant in terms of biomechanical properties. Indeed, modulating EMT transcription factors or epithelial cell adhesion molecule (EpCAM) not only affected cell migration and invasion, but also drastically modulated intracellular stiffness [115,116].

## 7. Conclusions

Despite the great effort made in searching for new molecular biomarkers for cancer, classification, stratification and prediction of outcome remains difficult. Taking into account the tumoral microenvironment seems to be crucial for understanding tumor transformation and progression. Numerous studies on the immune microenvironment have been conducted in recent years, but the morphological and biomechanical properties of the tumor in its surrounding environment have been less studied. However, there is an increasing development of high throughput techniques associating, for instance, the measurement of nanomechanical parameters in microfluidics and improved microscopy technologies combined with 3D biological models. Thus, in the future, biomechanics will be more and more easy to study and could help to better understand cancer initiation, progression and dissemination, and even cancer therapy resistance, since stiffness modulation has also been linked to drug resistance [117,118,119].

## Figures and Tables

**Figure 1 cells-10-00887-f001:**
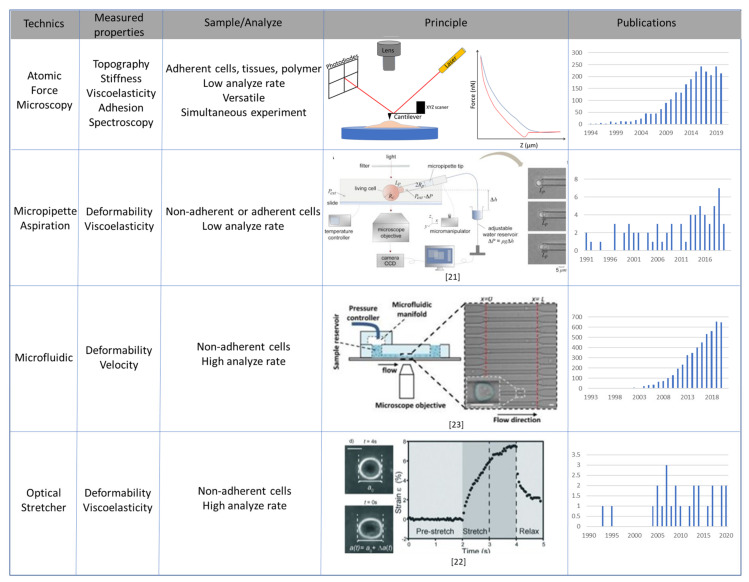
Overview of main techniques used to study cell or tissue biomechanical properties.

**Figure 2 cells-10-00887-f002:**
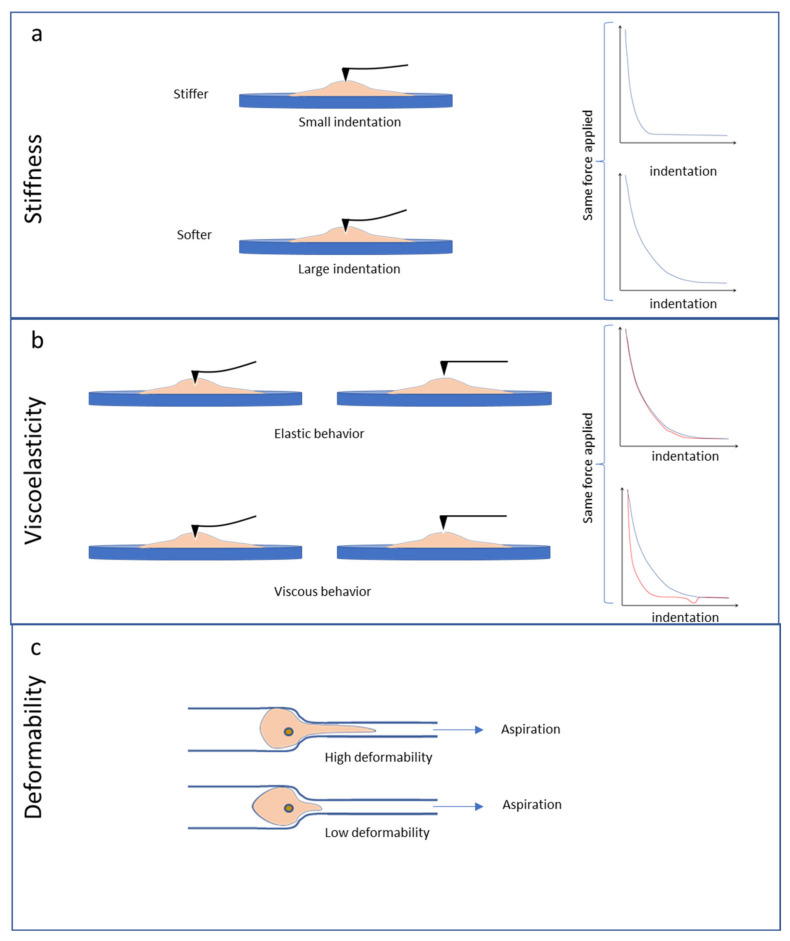
Schematic representation of cell stiffness, viscoelasticity and deformability properties. (**a**) For a given applied force, a larger indentation is observed for the softer cell. (**b**) A cell quickly recovers its original shape after indentation with small hysteresis for a more pronounced elastic behavior. Larger hysteresis is observed for a more viscous behavior. (**c**) The capacity of the cell to penetrate a thin capillary allows its deformability to be determined.

## Data Availability

No new data were created or analyzed in this study. Data sharing is not applicable to this article.

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
