# Peer review of "Biomechanical Properties of Cancer Cells"

_cells, 2021, doi:10.3390/cells10040887_

Round 1
Reviewer 1 Report
The authors of this review manuscript explore an important but poorly understood link between cellular biomechanics and tumor development.
Overall, the manuscript is timely, succinct, and logically constructed. However, there are several concerns that need to be addressed before the manuscript can be considered for publication.
1. Major concerns:
1.1. The authors provide a nice overview of the main methods to study biomechanical properties of cells and imply that these methods have a great potential “to better understand cancer initiation, progression and dissemination, and even cancer therapy resistance”. However, in order for this idea to materialize, it is essential to understand how the mechanisms underlying cancer development translate into the change of cellular biomechanics and vice versa, i.e. how the mechanical cues translate into specific biochemical cues (the process termed mechanotransduction). In this regard, the authors focus primarily on the role of the cytoskeleton, while omitting other aspects of the control of cellular biomechanics. The manuscript would benefit from a broader overview of the interface between cellular biochemistry and biomechanics (e.g. doi.org/10.1083/jcb.201701039 and doi:10.1038/s41556-017-0012-0).
1.2. The English language should be improved throughout the article. Some sentences are too long, ambiguous and difficult to understand (e.g. see below, minor points).
1.3. It would be more logical and easier to read the paper if the order of technics in Figure 1 corresponds to the description of these technics in section 2. This section begins with “2.1. Atomic Force Microscopy”, a technique that is in the row 2 (and not in row 1) of Figure 1.
1.4. The schematic panels in the column “Principle” of Figure/Table 1 are too small; the text is nearly impossible to read. I suggest to either increase the size of these schematic panels (the column “Principle” can be widened) or to illustrate the schematic principles in a separate figure.
2. Minor points:
2.1. The authors should check all abbreviations and make sure that they are uniform and deciphered only once, at the first mentioning. This is not the case for some abbreviations. For example, line 227: “deformability cytometry (DC)”, but on line 214 “real-time deformation cytometry (RT-DC)”. Line 230: “circulating tumors cells (CTCs)”. “Circulating Tumor Cells” have already been abbreviated on line 36.
2.2. Figure 1, column 2. “Measured properties” is more appropriate.
2.3. Lines 81-82: “The AFM is a type of scanning probe microscope”. The AFM (i.e. Atomic Force Microscopy) cannot be a microscope. Please reformulate the sentence.
2.4. Line 84: “More precisely, the AFM principle is based on a very small tip…”.
2.5. Line 126: “of a glass micropipette that forces the cell to remodel its shape”?
2.6. Lines 138-139: “a small chip”?
2.7. Line 148: Perhaps “description” can be replaced by “properties” or skipped altogether.
2.8. Line 236: “cell cytoskeleton” is a pleonasm.
2.9. Lines 239-240: “Concentration variation of the different molecules that compose the cytoskeleton influences its mechanical properties and its capacity to deform…” - It’s not only the concentration variation of the cytoskeleton-composing molecules but also their compartmentalization, posttranslational modifications etc.
2.10. Lines 260-261: “disrupting actin network and/or microtubules polymerization by drugs such as cytochalasins [61] or docetaxel led to cell nanomechanical changes [62–67]”. Docetaxel does not disrupt microtubule polymerization. To the contrary, it is a microtubule-stabilizing agent.
2.11. Line 281: “latter” seems to be a misspelling; it should be “later”.
2.12. Lines 283-285: “Paclitaxel treatment, which inhibits microtubule assembly, has been shown to decrease cell stiffness in Ishikawa and Hela cells [72] but to increase Young’s modulus in melanoma cells [73].” Paclitaxel (Taxol) and docetaxel (Taxotere) do not inhibit microtubule assembly. To the contrary, both are inhibitors of microtubule depolymerization, i.e. the microtubule-stabilizing drugs [see comment 2.10]. Also, please change to “in Ishikawa cells and HeLa cells”.
2.13. Lines 285-288: “Colchicin or docetaxel drugs, that induces functional disturbances in microtubule polymerization by blocking microtubule dynamics, seems to increase the elastic modulus in prostate cancer cells [74] and in some hepatoma cells, without affecting normal hepatocytes [75].”
-The correct name of the drug is “Colchicine”.
-It is unclear to which of the two drugs—colchicine or docetaxel—the verb “induces” refers to. Colchicine is an inhibitor of microtubule polymerization, whereas docetaxel is an inhibitor of microtubule depolymerization.
2.14 Line 303: “In fibroblasts and breast cells…”. The breast also contains fibroblasts. Please specify which breast cells. The breast epithelial cells?
2.15 Lines 319-320: “The importance of Ras/Rock GTPases cytoskeleton organization in stiffness modulation was also shown in epithelial transformed cells [48].” An unclear sentence. “The importance of Ras/Rock GTPase-dependent cytoskeleton organization…”?
2.16 Lines 329-332: “Most studies focused on nanomechanical properties of isolated cancer cells and only recently were analyzed the stiffness, deformability and morphology features of cancer cells in three dimensions, integrating the important role of tissue structure and surrounding extracellular matrix (ECM).” A problem with sentence construction? Also, add references supporting this statement.
2.17 Lines 340-343: “The interplay between cellular and ECM stiffness was all the more crucial that it exists a mechanoreciprocity between ECM and tumor in expansion inside the surrounding normal tissue that could be different for cells at the periphery versus the core of the tumor.” An unclear sentence.
2.18 Line 400: Decipher what is EpCAM.
Reviewer 2 Report
In view of the importance of biomechanical properties of cancer cells in carcinogenesis, the review by Runel et al is timely. However, the manuscript will be greatly improved if the review was written in a language that will make sense to readers of cells.
It is not entirely clear what defines the biomechanics properties of cancer cells and what their effects on the various hallmarks of cancer. The description was mostly on cultured cancer cells either as monoculture or cancer cells cultured ex vivo after isolated from tumor. It is not completely obvious of how the same techniques are used to measure the biomechanical properties in cancer cells existed in the context of tumor Microenvironment.
Other comments include:
In line 248-9 it is not clear what concentrations were referred to? It is also not clear how to measure and determine when filament is broken?
In line 372 what is finite element analysis?
Round 2
Reviewer 1 Report
The authors have addressed my criticism, although they took a minimalist approach in addressing the comment 1.1. Overall, I am satisfied with the revised manuscript, and I have only a few minor suggestions as follows:
-Line 42: delete "been".
-Line 69: delete "of" - "increased cell motility".
-Line 69-70: change to "(reviewed in [6])".
-Line 95: delete the whole line.
-Line 165: replace "in" with "on" ("4.1. Pioneering experiments on the biomechanics in cancer cells).
-Line 220: replace "were" with "was".
-Line 245: delete the second word "cell", which is a pleonasm ("is unquestionably the cytoskeleton").
Author Response
The authors have addressed my criticism, although they took a minimalist approach in addressing the comment 1.1. Overall, I am satisfied with the revised manuscript, and I have only a few minor suggestions as follows:
-Line 42: delete "been".
-Line 69: delete "of" - "increased cell motility".
-Line 69-70: change to "(reviewed in [6])".
-Line 95: delete the whole line.
-Line 165: replace "in" with "on" ("4.1. Pioneering experiments on the biomechanics in cancer cells).
-Line 220: replace "were" with "was".
-Line 245: delete the second word "cell", which is a pleonasm ("is unquestionably the cytoskeleton").
We thank Reviewer #1 for his positive comments on this new version of our manuscript and we have addressed all of his minor comments in this new version of our manuscript.
Reviewer 2 Report
It is clear that there are changes in biomechanics properties in both cancer cells and tumor microenvironment. There was little in this review discussing the interplay between them. It is also not exactly what are the altered biomechanics properties and how do the changes affect cancer hallmarks? It is also not obvious from the review if changes in biomechanics properties in either driver or passenger of the carcinogenesis process.
Author Response
We apologize to not completely answer the questions raised by Reviewer #2 by the data presented in our review. It is clearly difficult to answer the question of the biomechanics properties as either driver or passenger of tumorigenesis or to discuss the interplay between cancer cells and their surrounding microenvironment. Indeed, studies that have been conducted for the moment just started to answer these questions.